# Demographic and Lifestyle Factors and Memory in European Older People

**DOI:** 10.3390/ijerph16234727

**Published:** 2019-11-27

**Authors:** Michal Steffl, Tereza Jandova, Klara Dadova, Iva Holmerova, Piergiusto Vitulli, Sante D. Pierdomenico, Tiziana Pietrangelo

**Affiliations:** 1Faculty of Physical Education and Sport, Charles University, 162 52 Prague, Czech Republic; dadova.klara@gmail.com; 2Department of Neuroscience, Imaging and Clinical Science, Università degli Studi G. d‘Annunzio Chieti e Pescara, 66100 Chieti, Italy; te.jandova@hotmail.com (T.J.); tiziana@unich.it (T.P.); 3Faculty of Humanities, Charles University, 158 00 Prague, Czech Republic; iva.holmerova@gerontocentrum.cz; 4Department of Medical, Oral and Biotechnological Sciences, Università degli Studi G. d‘Annunzio Chieti e Pescara, 66100 Chieti, Italy; piergiusto@gmail.com (P.V.); pierdom@unich.it (S.D.P.)

**Keywords:** demographic and lifestyle factors, ageing, word-list learning test, memory

## Abstract

Objectives: To investigate associations between demographic and lifestyle factors and memory performance in European people aged ≥60 years. Methods: Data from 23,641 people with a mean age of 70.2 (95 % CI 70.1–70.3) were analyzed and drawn from the fourth wave of the Survey of Health, Ageing, and Retirement in Europe (SHARE). Generalized linear models were carried out to estimate the associations for both men and women. Memory performance was tested using two word-list learning tests with immediate and delayed recall in SHARE. Results: age, severe limitations in physical activities, and any past alcohol problem were all negatively associated with memory performance. Contrarily, education level, higher nonalcoholic fluid intake, and engagement in sports activities more than once a week and in activities requiring a moderate level of energy were all positively associated with memory performance. Smoking showed a significant negative association only in the immediate recall test for both men and women together, whilst long-term illness showed association only in the delayed recall. Alcohol consumption was positively associated with memory performance in women, but in men, it depended on the drinking frequency. Conclusions: Demographic and lifestyle factors are associated with memory performance in the older population.

## 1. Introduction

Memory problems are among the most common complaints of the older adult population. Memory can be defined as the ability to acquire, process, store, and retrieve information and generally can be categorized into three main stages: sensory information store, short-term episodic memory, and long-term memory [1], which is supported by distinct brain regions [2]. One of the most prominent aspects of ageing is that memory processes show a decline, and many theories were proposed over the years to account for such observed deficits in memory function and ageing [3]. For example, combined findings of a recent study [4] have shown that age is associated with reduced grey matter volume and memory performance, even when allowing for educational differences, which have been long known to be associated with memory performance [5,6]. Moreover, memory impairments have been also found to be associated with common age-related diseases such as heart failure, diabetes, or cancer [7]. However, other factors related more to an individual’s lifestyle such as good hydration status, smoking, alcohol consumption, and physical activity have been suggested to play a major role [8,9,10].

For example, smoking and alcohol consumption have been both associated with cortical grey matter loss, memory impairments, and worse cognition [11,12,13]. On the other hand, physical activity was found to improve memory performance in both young and older people [14]. Numerous studies have also found negative associations between mild dehydration and several important aspects of cognitive function including short-term memory [15,16,17,18].

Therefore, whilst certain demographic factors cannot be changed, investigating their possible associations with memory can help to better establish and clarify the age-related memory decline. On the other hand, maintaining one’s health and avoiding unhealthy lifestyle habits may aid in the prevention or delay of such age-related memory decline. Hence, the main aim of this study was to investigate the associations between the above-outlined lifestyle factors and memory performance, along with health-related and demographic factors such as age and education level in cognitively healthy European people aged ≥60 years.

## 2. Materials and Methods

### 2.1. Study Design

This study was a cross-sectional study that analyzed data obtained from participants of the Survey of Health, Ageing, and Retirement in Europe (SHARE) Wave 4. SHARE is a multidisciplinary and cross-national panel database of microdata on health, socioeconomic status, and social and family networks of approximately 140,000 individuals aged 50 or older (around 380,000 interviews) where the data are obtained through questionnaires created by social and medical specialists—see Borsch-Supan et al. [19] for methodological details. The procedure (Waves 1–4 SHARE tests) was reviewed and approved by the Ethics Committee of the University of Mannheim, and Wave 4 of SHARE and the continuation of the project was reviewed and approved by the Ethics Council of the Max Planck Society [20].

### 2.2. Study Population

Data from people across 16 countries in Europe aged ≥60 who participated in the fourth wave of SHARE project were included in the analyses. To avoid inclusion of people with memory disorders (n = 214), we excluded participants who responded “Yes” for the questions: “Has a doctor ever told you that you had/Do you currently have any of the conditions on this card? Alzheimer’s disease, dementia, organic brain syndrome, senility, or any other serious memory impairment.” For this study, we also divided participants according to sex [21].

### 2.3. Education Level

Level of education has been suggested to play an important role in memory [5]; therefore, we included it as one of the factors in the analyses. For education level categorization, the International Standard Classification of Education (ISCED-97) [22] in SHARE was used.

### 2.4. Memory

Memory was tested using a standard version of two word-list learning tests [23] with immediate and delayed recall. For the tests, interviewers read 10 words and then asked the participant to recall as many of the words as possible, in any order. Participants were tested immediately and after approximately ten minutes.

### 2.5. Health-Related Factors

Health-related factors of participants were assessed by choosing long-term illnesses and limitation in activities obtained by the questions: “Do you have any long-term health problems, illness, disability, or infirmity?” with yes or no answers and “For the past six months at least, to what extent have you been limited because of a health problem in activities people usually do?” with three answer options: severely limited; limited, but not severely; and not limited.

### 2.6. Lifestyle Risk Factors

Information about daily fluid intake was obtained by the following question: “On a regular day, how many cups or glasses of tea, coffee, water, milk, fruit juice, or soft drinks in total do you normally drink?” with the following three options: 1–2 cups, 3–5 cups, and 6 cups or more. A cup is considered 200–240 mL in the SHARE. Smoking status was obtained by the question “Do you smoke at the present time?” with yes or no answers. Alcohol consumption was obtained by the question “During the last 3 months, how often did you drink any alcoholic beverages, like beer, cider, wine, spirits, or cocktails?” with the following answer options: Almost every day; five or six days a week; three or four days a week; once or twice a week; once or twice a month; and not at all or less than once a month. Moreover, any drinking problem was assessed by the question: “Was excessive drinking a problem at any time of your life?” with yes or no answers. About physical activities, participants were asked two questions: “How often do you engage in physical activity, such as sports, heavy housework, or a job that involves physical labour?” and “How often do you engage in activities that require a moderate level of energy such as gardening, cleaning the car, or taking a walk?” with the following answer options: more than once a week; once a week; one to three times a month; and hardly ever or never.

### 2.7. Statistical Analysis

First, descriptive statistics of participants were produced with calculated means and 95% confidence intervals (95% CI) for continuous variables and percentage and 95% CI for categorical variables. Then, generalized linear models for ordinal response were carried out to estimate the associations between the demographic and lifestyle factors age and performances in verbal memory tests for men, women, and both sexes. We used the Bonferroni correction to counteract the problem of multiple comparisons. To calculate the corrected CI, we multiplied a standard error (SE) by the Z-score = 2.807, which corresponded to the 10 individual hypotheses that were tested at a significance level of α/m where m = number of hypotheses. A statistically significant result, at the level of 99.5%, was when the CI did not cross the zero line. Data were analyzed using IBM SPSS Version 24.0 (SPSS Inc.; Chicago, IL, USA).

## 3. Results

Data from 23,641 older people (12,463 men and 11,178 women) with a mean age of 70.2 (95% CI 70.1–70.3) were analyzed. Women performed better in both memory tests (mean score 5.35 (95% CI 5.32–5.38) in the immediate recall and 3.98 (95% 3.94–4.02) in the delayed recall) when compared to men (mean score 4.94 (95% 4.91–4.97) in the immediate recall and 3.47 (95% 3.43–3.50) in the delayed recall). However, women were significantly younger than men. On the other hand, more men reported smoking, alcohol drinking, and drinking problems. Men were also more active in sports activities than women; nevertheless, there was no difference between men and women in physical activities requiring a moderate level of energy. Descriptive statistics are presented in Table 1.

According to the generalized linear models, there were several factors significantly associated with memory performance in both tests. Age and severe limitations in physical activities were negatively associated with memory performance in all the analyses regardless of sex difference. Any past alcohol problem related to excessive drinking was also negatively associated with memory in all the analyses except in women in the delayed recall. Smoking was associated negatively only in the immediate recall test for both men and women together, whilst long-term illness was associated only in the delayed recall. On the other hand, education level, sports activities, and activities requiring a moderate level of energy—more than once a week—were all positively associated with memory performance in all the analyses. Lastly, nonalcoholic fluid intake of 6 cups or more was positively associated with memory performance except in women in the delayed recall. Results of the generalized linear models are presented in Table 2 and Table 3.

## 4. Discussion

The main objective of this study was to investigate the associations between demographic and lifestyle factors and memory in the older adult population as all these factors have lately been recognized as important correlates (either positive or negative) of cognitive function including memory in the elderly [2,9,10,13,24,25,26]. The present study confirms and extends such findings by demonstrating that (1) age; (2) education level; (3) long term illness and severe limitation; (4) smoking; (5) alcohol consumption and any past excessive alcohol drinking problems; (6) nonalcoholic daily fluid intake; and (7) engagement in sport and physical activities are all associated with memory performance in the elderly population.

In this large sample of cognitively healthy European people aged ≥60 years, factors including age, severe limitation, and any past excessive alcohol drinking problems were significantly associated with worse performance in the two memory tests for both men and women (the exception was in women in the delayed recall). On the contrary, factors such as education level, higher daily nonalcoholic fluid intake (especially 6 cups and more), and engagement in sports activities once a week or more and in activities requiring a moderate level of energy were all significantly associated with better memory performance in both tests. Smoking had a significant negative association only in the immediate recall test for both men and women together, whilst long-term illness had a negative association only in the delayed recall. Alcohol consumption was associated with significant positive memory performance in both tests in women, whilst in men, alcohol consumption was positively associated only in the delayed recall, depending on the reported drinking frequency. Overall, these findings are consistent with past studies that examined similar associations; however, they warrant brief discussion in the context of several study limitations.

Starting with the demographics, overall, women performed better in both memory tests than men, which is in concordance with the already well-established phenomenon [21,27] (although, of note, the opposite is true for the demented population of elders [28]). However, another reason for this difference between men and women could be also the fact that women were significantly younger than men in our study. The factor of more advanced age is another well-known negative correlate of cognition including memory [29], and the results of our study further support this notion. Moreover, memory is not only vulnerable to ageing but also to disease states [7]. In our study, we tried to assess the health status of the participants by asking about any long-term illness and limitation in activities they (may) have, as it could potentially influence the memory performance in the tests. The results of our study show that severe limitation was associated with significant negative memory performance in both tests, whilst long term illness was associated only in the delayed recall, thus confirming and extending previous findings [7]. For example, a typical finding of population-based and clinical-based studies is that many diseases affect different types of memory differently [7]; however, our study design, as it is, cannot draw such conclusion but it may indicate to some trends that could be investigated by future studies. In terms of the education level, there were no statistical differences between men and women and our regression model has shown a positive association of education level and memory performance for both sex, i.e., the higher the education level, the better the score. Again, this result of our study is consistent with findings of other studies [30] that found positive association of education level with memory performance, which brings attention to the necessity of considering the level of education when interpreting the results of memory tests, specifically when detecting, for example, cognitive impairment such as dementia or Alzheimer’s disease.

Moving towards the lifestyle factors, there were more men than women who self-reported smoking, drinking more alcohol, and having some past alcohol drinking problems, which could be another reason why females scored better in both memory tests. It has been previously found that persistent smoking leads to impairments in everyday prospective memory [13] or that smokers score lower than nonsmokers in global cognitive function [31]; therefore, the negative association of smoking with memory performance found in our study supports these prior claims. Moreover, a recent study by Marshall et. al. [26] provided evidence that combined excessive drinking and cigarette smoking led to greater impairment in time-based prospective memory, which is, in fact, fundamental for everyday independent living. Other studies have also found negative associations between alcohol consumption and memory. For instance, results from a 19-year prospective cohort study showed that middle-aged adults with a history of alcohol use disorders have increased odds of developing severe memory impairment later in life [9]. Again, the latter finding is in line with our results that also show a strong negative association of self-reported past drinking problems with memory performance. Similarly, another recent study has also suggested that heavy current alcohol consumption is associated with significant impairment in several neurocognitive domains such as learning, memory, or motor function with lasting negative consequences [10].

However, more intrigued are our results about alcohol consumption as they were found to be positively associated with memory performance in both tests, depending on sex and self-reported frequency. In women, alcohol consumption was positively associated with memory performance in both tests. This finding is consistent with the past study where moderate female drinkers who drank less than 15.0 g of alcohol per day, which corresponds more or less to one standard drink [32], had better mean cognitive scores than nondrinkers [33], however, such benefits were not seen in older men [34]. Our results show a similar pattern, but the results of the association of alcohol consumption with memory performance for men are inconsistent. Therefore, the health benefits of moderate drinking for memory are still questionable, and if they exist, they are probably limited to a small amount of alcohol as suggested by a recent clinical review [35]. Furthermore, it is also possible that such results could be partly related to the positive association of higher nonalcoholic fluid intake and memory performance, which was also found in our study.

Numerous studies have found negative associations between mild dehydration and several important aspects of cognitive function including short-term memory [15,36]. There is evidence that being dehydrated by just 2% leads to impaired performance in tasks that require attention, psychomotor, and immediate memory skills [37,38]. However, a study conducted by Young and Benton [39] has shown that even 0.6% loss of body mass reduced the efficiency of working memory. The results of our study also showed a positive association of higher (specifically 6 cups and more) nonalcoholic fluid intake with memory performance in both tests for both men and women. However, it is also possible that women who self-reported to drink alcohol could have, in fact, even higher fluid intake than men as they are more likely to incline to drink beverages with higher content of water such as long drinks, whilst men are more likely to choose stronger types of alcoholic drinks such as shots [40], which could result in better hydration status of women and, as a consequence, further influencing the memory test performance (the same could be true for the men too, indeed). On the other hand, although women generally consume less alcohol compared to men, they usually suffer more severe brain and other organ damage following binge or chronic alcohol abuse [41]. Nevertheless, our study could not precisely calculate or estimate the exact level of hydration (as otherwise measured by bioimpedance or by the loss of body mass) and/or liquid consumption of both alcoholic and nonalcoholic drinks, but it certainly can highlight the importance of good hydration among elderly for any future investigations as older adults are particularly susceptible to mild dehydration as the total body water decreases with ageing [38], largely as a result of compromised homeostatic mechanisms such as the loss of thirst sensation and decline in renal concentrating capacity or as a result of sarcopenia, which causes a smaller fluid reserve [18,36,37,42,43].

In terms of physical activity, men were more active in sports activities than women; however, there was no significant difference between them in physical activities requiring a moderate level of energy. Such engagement in sports activities (once a week or more) and in activities requiring a moderate level of energy appeared to be associated with a more significant positive memory performance in both tests, which is in concordance with the findings of recent studies that have shown that aerobic exercise, in particular, is of benefit to both young and older adults improving memory function [44,45], even independently of its intensity [46]. In contrast, it has been found that lower physical activity predicted an acute decline in attention/executive function in heart failure patients at a 12-month follow-up and decreased cerebral perfusion, which as a consequence, may exacerbate the risk for Alzheimer’s disease and worsen cognitive function [47,48]. Therefore, contrary to the moderate alcohol consumption paradigm, the positive effect of physical activity on memory is beyond any doubt.

In summary, whilst it is certainly valuable to find different lifestyle and/or demographic factors that influence memory in old age, it is important to remember that our data are cross sectional and taken from SHARE, which makes it difficult to draw any causation per se or to distinguish true effects of such factors from cohort effect. Despite SHARE’s unique multidisciplinary and cross-national panel data database (ex ante harmonized with many validated methods [19]), which has already allowed for numerous comparisons in a longitudinal setting, some concerns about using SHARE have been noted. In particular, to inform about public health due to the noted low response rates and moderate levels of attrition, which can potentially generate sample selection bias, can limit the representativeness of the database and the generalizability of results (even despite the provision of calibrated weights). In addition, there may have been sampling and other biases in the available participants: for example, majority of our data were based on self-reports, which are known to be of varying reliability and validity such as an awareness of one’s condition, specifically of the early stage of dementia or Alzheimer disease is known to vary among people. However, biases could be also introduced due to the data collection itself as SHARE interviewers used computer-assisted personal interviewing (CAPI) to collect most of the data, self-administered questionnaires (drop-off) after completion of the CAPI, and telephone interviews (CATI) with a proxy, conducted on behalf of some incapable participants [19].

Furthermore, the inability to precisely quantify or estimate some parameters such as in the case of nonalcoholic or alcoholic fluid intake makes it difficult to clarify the findings. Therefore, all these factors could have obscured the true associations between the studied factors, thus increasing the uncertainty of the resulting generalizations. Nevertheless, our findings emphasize the possible influence of lifestyle and demographic factors on memory performance, and despite its limitations, the findings of the present study are very consistent with the majority of the past studies discussed above. Our findings also highlight the importance of evaluating sex-specific differences in memory function, specifically based on lifestyle behaviour.

In conclusion, the results of our regression models showed a link between demographic and lifestyle factors and performance in memory tests in older individuals. Our study, despite being based on a cross-sectional data taken from SHARE, could have practical implications such as to promote a healthy lifestyle among the older population; however, confirmation that these factors truly influence memory should be investigated by longitudinal epidemiological studies in the future.

## Figures and Tables

**Table 1 ijerph-16-04727-t001:** Participant population of the current study.

	Together (n = 23,641)	Men (n = 12,463)	Women (n = 11,178)
*Age (years)*	70.2 (7.1–7.3)	70.3 (70.2–70.4)	70.1 (70.0–70.1)
*Memory*			
Immediate recall (words)	5.13 (5.11–5.15)	4.94 (4.91–4.97)	5.35 (5.32–5.38)
Delayed recall (words)	3.71 (3.68–3.74)	3.47 (3.43–3.50)	3.98 (3.94–4.02)
*ISCED 1997*			
Second stage of tertiary education	0.9 (0.8–0.11)	1.1 (1.0–1.3)	0.7 (0.6–0.9)
First stage of tertiary education	21.3 (20.8–21.8)	22.8 (22.1–23.6)	19.6 (18.9–20.3)
Post-secondary non-tertiary education	4.6 (4.4–4.9)	4.3 (4.0–4.7)	5.0 (4.6–5.4)
Upper secondary education	33.7 (33.1–34.3)	34.3 (33.4–35.1)	33.1 (32.3–34.0)
Lower secondary education or second stage of basic education	17.8 (17.3–18.3)	16.5 (16.5–17.8)	19.1 (18.4–19.9)
Primary education or first stage of basic education	19.3 (18.8–19.8)	18.5 (17.8–19.2)	20.1 (19.4–20.9)
Pre-primary education	2.3 (2.1–2.5)	2.4 (2.2–2.7)	2.2 (2.0–2.5)
*Long-term illness*	52.2 (51.6–52.9)	52.2 (51.4–53.1)	52.3 (51.3–53.2)
*Limited activities*			
Severely limited	13.4 (12.9–13.8)	13.2 (12.6–13.8)	13.5 (12.9–14.2)
Limited, but not severely	34.1 (33.5–34.7)	32.7 (31.9–33.6)	35.5 (34.6–36.4)
Not limited	52.6 (51.9–53.2)	54.0 (53.2–54.9)	50.9 (50.0–51.9)
*Daily nonalcoholic fluid intake*			
6 cups or more	62.6 (62.0–63.2)	62.1 (61.2–62.9)	63.2 (62.3–64.1)
3–5 cups	32.4 (31.8–33.0)	32.4 (31.6–33.2)	32.5 (31.6–33.4)
1–2 cups	5.0 (4.7–5.2)	5.5 (5.1–6.0)	4.3 (4.0–4.7)
*Smoking at the present time*	16.3 (15.8–16.7)	19.2 (18.5–19.9)	13.0 (12.4–13.7)
*Days a week alcohol consumption last 3 months*			
Almost every day	30.3 (29.8–30.9)	38.8 (37.9–39.7)	20.9 (20.2–21.7)
Five or six days a week	4.1 (3.9–4.4)	4.9 (4.5–5.3)	3.3 (2.9–3.6)
Three or four days a week	9.0 (8.7–9.4)	10.4 (9.8–10.9)	7.6 (7.1–8.1)
Once or twice a week	23.3 (22.8–23.8)	23.2 (22.5–24.0)	23.4 (22.6–24.2)
Once or twice a month	16.5 (16.0–17.0)	12.7 (12.1–13.3)	20.7 (20.0–21.5)
Not at all or less than once a month	16.7 (16.2–17.2)	10.1 (9.5–10.6)	24.1 (23.4–25.0)
*Drinking problems*	3.6 (3.3–3.8)	5.4 (5.0–5.8)	1.5 (1.3–1.7)
*Sports or activities that are vigorous*			
More than once a week	33.2 (32.6–33.8)	36.6 (35.8–37.5)	28.8 (28.0–29.7)
Once a week	14.8 (14.3–15.2)	14.1 (13.5–14.7)	15.3 (14.6–16.0)
One to three times a month	9.3 (8.9–9.6)	9.6 (9.1–10.1)	8.7 (8.2–9.3)
Hardly ever, or never	43.6 (42.9–44.2)	39.7 (38.9–40.6)	47.1 (46.2–48.1)
*Activities requiring a moderate level of energy*			
More than once a week	72.0 (71.4–72.5)	72.1 (71.4–72.9)	71.7 (70.9–72.6)
Once a week	13.5 (13.1–13.9)	13.3 (12.7–13.9)	13.7 (13.1–14.4)
One to three times a month	5.4 (5.1–5.6)	5.7 (5.3–6.1)	5.0 (4.6–5.4)
Hardly ever or never	9.2 (8.8–9.6)	8.9 (8.4–9.4)	9.5 (9.0–10.1)

Note: Values are expressed as means (95% CI) in continuous or percentage (95% CI) in categorical variables.

**Table 2 ijerph-16-04727-t002:** Generalized linear models for the immediate recall.

	Together Est.	Men Est.	Women Est.
*Age*	**−0.059 (−0.063–−0.055)**	**−0.058 (−0.064–−0.052)**	**−0.057 (−0.063–−0.051)**
*ISCED 1997*			
Second stage of tertiary education	**2.089 (1.776–2.402)**	**2.020 (1.580–2.460)**	**2.317 (1.783–2.851)**
First stage of tertiary education	**1.891 (1.713–2.069)**	**1.824 (1.560–2.088)**	**2.032 (1.752–2.312)**
Post-secondary non-tertiary education	**1.631 (1.424–1.838)**	**1.634 (1.320–1.948)**	**1.669 (1.350–1.988)**
Upper secondary education	**1.416 (1.241–1.591)**	**1.325 (1.065–1.585)**	**1.570 (1.297–1.843)**
Lower secondary education or second stage of basic education	**1.060 (0.881–1.239)**	**0.989 (0.720–1.258)**	**1.149 (0.871–1.427)**
Primary education or first stage of basic education	**0.618 (0.441–0.795)**	**0.578 (0.313–0.843)**	**0.671 (0.395–0.947)**
Pre-primary education	0 ^a^	0 ^a^	0 ^a^
*Long-term illness*			
Yes	−0.035 (−0.096–0.026)	0.020 (−0.072–0.112)	−0.051 (−0.145–0.043)
No	0^a^	0^a^	0^a^
*Limited activities*			
Severely limited	**−0.221 (−0.313–−0.129)**	**−0.268 (−0.409–−0.127)**	**−0.180 (−0.320–−0.040)**
Limited, but not severely	−0.056 (−0.120–0.008)	−0.60 (−0.157–0.037)	−0.076 (−0.174–0.022)
Not limited	0 ^a^	0 ^a^	0 ^a^
*Daily nonalcoholic fluid intake*			
6 cups or more	**0.329 (0.208–0.450)**	**0.257 (0.083–0.431)**	**0.365 (0.168–0.562)**
3–5 cups	**0.164 (0.040–0.288)**	**0.109 (−0.070–0.288)**	**0.206 (0.004–0.408)**
1–2 cups	0 ^a^	0 ^a^	0 ^a^
*Smoking at the present time*			
Yes	**−0.095 (−0.166–−0.024)**	**−0.094 (−0.195–0.007)**	**0.005 (−0.113–0.123)**
No	0 ^a^	0 ^a^	0 ^a^
*Days a week alcohol consumption last 3 months*			
Almost every day	−0.065 (−0.144–0.014)	0.013 (−0.125–0.151)	0.115 (−0.004–0.234)
Five or six days a week	0.070 (−0.071–0.211)	0.201 (−0.013–0.415)	0.102 (−0.131–0.335)
Three or four days a week	0.063 (−0.043–0.169)	0.069 (−0.103–0.241)	**0.279 (0.114–0.444)**
Once or twice a week	0.071 (−0.012–0.154)	0.065 (−0.082–0.212)	**0.215 (0.100–0.330)**
Once or twice a month	0.045 (−0.014–0.123)	−0.060 (−0.223–0.103)	**0.158 (0.040–0.276)**
Not at all or less than once a month	0 ^a^	0 ^a^	0 ^a^
*Drinking problems*			
Yes	**−0.456 (−0.596–−0.316)**	**−0.333 (−0.506–−0.160)**	**−0.376 (−0.704–−0.048)**
No	0 ^a^	0 ^a^	0 ^a^
*Sports or activities that are vigorous*			
More than once a week	**0.098 (0.033–0.163)**	**0.155 (0.055–0.255)**	**0.139 (0.038–0.240)**
Once a week	**0.102 (0.021–0.183)**	0.123 (−0.003–0.249)	0.106 (−0.014–0.226)
One to three times a month	0.024 (−0.071–0.119)	0.067 (−0.076–0.210)	0.051 (−0.096–0.198)
Hardly ever, or never	0 ^a^	0 ^a^	0 ^a^
*Activities requiring a moderate level of energy*			
More than once a week	**0.350 (0.251–0.449)**	**0.367 (0.214–0.520)**	**0.314 (0.165–0.463)**
Once a week	**0.210 (0.095–0.325)**	**0.237 (0.060–0.414)**	**0.184 (0.011–0.357)**
One to three times a month	**0.183 (0.041–0.325)**	**0.247 (0.035–0.459)**	0.164 (−0.056–0.384)
Hardly ever or never	0 ^a^	0 ^a^	0 ^a^

Note: Est. = Unstandardized coefficient B (99.5 % CI); ^a^ set to zero because this parameter is redundant (reference value); values in bold indicate statistically significant results.

**Table 3 ijerph-16-04727-t003:** Generalized linear models for the delayed recall.

	Together Est.	Men Est.	Women Est.
*Age*	**−0.069 (−0.074–−0.064)**	**−0.065 (−0.072–−0.058)**	**−0.070 (−0.077–−0.063)**
*ISCED 1997*			
Second stage of tertiary education	**2.362 (1.974–2.750)**	**2.132 (1.610–2.654)**	**2.845 (2.153–3.537)**
First stage of tertiary education	**2.123 (1.902–2.344)**	**1.947 (1.633–2.261)**	**2.403 (2.041–2.765)**
Post-secondary non-tertiary education	**1.765 (1.508–2.022)**	**1.622 (1.249–1.995)**	**1.970 (1.558–2.382)**
Upper secondary education	**1.506 (1.289–1.723)**	**1.327 (1.018–1.636)**	**1.775 (1.421–2.129)**
Lower secondary education or second stage of basic education	**1.016 (0.794–1.238)**	**0.823 (0.504–1.142)**	**1.248 (0.887–1.609)**
Primary education or first stage of basic education	**0.624 (0.404–0.844)**	**0.509 (0.195–0.823)**	**0.775 (0.417–1.133)**
Pre-primary education	0 ^a^	0 ^a^	0 ^a^
*Long-term illness*			
Yes	**−0.102 (−0.184–−0.020)**	−0.069 (−0.178–0.040)	−0.083 (−0.204–0.038)
No	0 ^a^	0 ^a^	0 ^a^
*Limited activities*			
Severely limited	**−0.222 (−0.346–−0.098)**	**−0.279 (−0.446–−0.112)**	**−0.181 (−0.362–−0.000)**
Limited, but not severely	**−0.088 (0.174–−0.002)**	−0.055 (−0.171–0.061)	**−0.158 (−0.285–−0.031)**
Not limited	0 ^a^	0 ^a^	0 ^a^
*Daily nonalcoholic fluid intake*			
6 cups or more	**0.298 (0.135–0.461)**	**0.283 (0.077–0.489)**	0.243 (−0.012–0.498)
3–5 cups	0.102 (−0.065–0.269)	0.110 (−0.102–0.322)	0.053 (−0.208–0.314)
1–2 cups	0 ^a^	0 ^a^	0 ^a^
*Smoking at the present time*			
Yes	−0.018 (−0.113–0.077)	−0.004 (−0.078–0.089)	0.097 (−0.056–0.250)
No	0 ^a^	0 ^a^	0 ^a^
*Days a week alcohol consumption last 3 months*			
Almost every day	−0.006 (−0.112–0.100)	0.060 (−0.104–0.224)	**0.246 (0.092–0.400)**
Five or six days a week	**0.221 (0.030–0.412)**	**0.303 (0.049–0.557)**	**0.367 (0.066–0.668)**
Three or four days a week	**0.155 (0.012–0.298)**	0.137 (−0.067–0.341)	**0.444 (0.230–0.658)**
Once or twice a week	**0.136 (0.025–0.247)**	0.110 (−0.064–0.283)	**0.324 (0.175–0.473)**
Once or twice a month	0.104 (−0.016–0.224)	-0.083 (−0.277–0.111)	**0.278 (0.125–0.431)**
Not at all or less than once a month	0 ^a^	0 ^a^	0 ^a^
*Drinking problems*			
Yes	**−0.582 (−0.770–−0.394)**	**−0.464 (−0.669–−0.259)**	−0.316 (−0.740–0.108)
No	0 ^a^	0 ^a^	0 ^a^
*Sports or activities that are vigorous*			
More than once a week	**0.146 (0.058–0.234)**	**0.203 (0.085–0.321)**	**0.227 (0.096–0.358)**
Once a week	**0.161 (0.052–0.270)**	**0.206 (0.056–0.356)**	0.149 (−0.006–0.304)
One to three times a month	0.039 (−0.089–0.167)	0.140 (−0.029–0.309)	0.018 (−0.173–0.209)
Hardly ever or never	0 ^a^	0 ^a^	0 ^a^
*Activities requiring a moderate level of energy*			
More than once a week	**0.298 (0.164–0.432)**	**0.264 (0.082–0.446)**	**0.301 (0.108–0.494)**
Once a week	**0.211 (0.056–0.366)**	0.186 (−0.025–0.397)	**0.230 (0.006–0.354)**
One to three times a month	0.158 (−0.033–0.349)	0.216 (−0.036–0.468)	0.143 (−0.142–0.428)
Hardly ever, or never	0 ^a^	0 ^a^	0 ^a^

Note: Est. = Unstandardized coefficient B (99.5 % CI); ^a^ set to zero because this parameter is redundant (reference value); values in bold indicate statistically significant results.

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
