# Peer review of "Demographic and Lifestyle Factors and Memory in European Older People"

_ijerph, 2019, doi:10.3390/ijerph16234727_

Round 1

Reviewer 1 Report

The title of the article would be better described as 'Fluid intake of non-alcoholic beverages and memory'. 

Please consider and discuss the role of alcohol intake in your introduction and results.

Line 160 The statement in the conclusions ' It can be concluded that according to our regression models a fluid intake of 6 and more cups (1200-1440ml) of non-alcoholic drinks a day improves memory performance,'  This needs to be changed to 'a fluid intake of 6 and more cups (1200-1440ml) of non-alcoholic drinks a day was associated with memory performance'

Author Response

Dear reviewer,

Thank you very much for reviewing our manuscript. We also greatly appreciate the insightful suggestions and helpful comments.

Our specific responses follow:

The title of the article would be better described as 'Fluid intake of non-alcoholic beverages and memory'.

Agreed upon and changed according to the reviewer’s suggestion.

Please consider and discuss the role of alcohol intake in your introduction and results.

The role of alcohol intake is now more, yet still briefly mentioned in the introduction, followed by appropriate references. It is also mentioned briefly in the discussion section – as another limitation of the study.

In the matter of fact, when we were preparing the manuscript at the beginning of the process, we had a whole section on the role of alcohol intake and memory. However, we all agreed that for the purpose of this article, it is not necessary to go to such extent as the article’s main aim is to draw attention to the importance of non-alholic fluid intake and memory.

Line 160 The statement in the conclusions ' It can be concluded that according to our regression models a fluid intake of 6 and more cups (1200-1440ml) of non-alcoholic drinks a day improves memory performance,'  This needs to be changed to 'a fluid intake of 6 and more cups (1200-1440ml) of non-alcoholic drinks a day was associated with memory performance'.

Agreed upon and changed according to the reviewer’s suggestion.

Reviewer 2 Report

The principal aim of this study was to investigate the association between self-reported daily fluid intake and memory performance in healthy active European people aged ≥ 60 years. The results of this study suggest that a fluid intake of ≥ 6 cups of non-alcoholic drink a day contributes to better memory performance in older population. The study is well-designed and well-conducted, and the results are interesting and substantial for the community. However, authors must rework their discussion to refine their article.

Comments:

In line 36, the authors cited factors that can influence memory. Among these factors, we have physical activity. To control this factor, the authors excluded participants based on a single question about physical activity practice (see lines 75-77). In my opinion, the authors should measure more seriously the level of physical activity using the measures recognized in the literature (using actimeter, physical activity questionnaire). I suggest that the authors write a paragraph in the discussion to clarify this limit of the study.

Author Response

Dear reviewer,

Thank you very much for reviewing our manuscript. We also greatly appreciate the insightful suggestions and helpful comments.

Our specific responses follow:

The principal aim of this study was to investigate the association between self-reported daily fluid intake and memory performance in healthy active European people aged ≥ 60 years. The results of this study suggest that a fluid intake of ≥ 6 cups of non-alcoholic drink a day contributes to better memory performance in older population. The study is well-designed and well-conducted, and the results are interesting and substantial for the community. However, authors must rework their discussion to refine their article.

Comments:

In line 36, the authors cited factors that can influence memory. Among these factors, we have physical activity. To control this factor, the authors excluded participants based on a single question about physical activity practice (see lines 75-77). In my opinion, the authors should measure more seriously the level of physical activity using the measures recognized in the literature (using actimeter, physical activity questionnaire). I suggest that the authors write a paragraph in the discussion to clarify this limit of the study.

Agreed upon and a few lines were added in the discussion section as another limitation of the study with a suggestion of how to control better this factor in future studies.

Reviewer 3 Report

It is a rare occasion to review a manuscript that requires no improvement, at least that I can see. Congratulations on your concise, clear and interesting paper. The results are plausible and reiterate the importance of adequate hydration for an important aspect of cognitive function in older people. 

The only suggestions I have is: i) Line 54: Sentence should read 'The majority of studies focusing on...'; and ii) that Table 2 be presented on the one page.

Congratulations on your work and thank you for the opportunity to review such a well presented study report. 

Author Response

Dear reviewer,

Thank you very much for reviewing our manuscript. We also greatly appreciate the insightful suggestions and helpful comments.

Our specific responses follow:

It is a rare occasion to review a manuscript that requires no improvement, at least that I can see. Congratulations on your concise, clear and interesting paper. The results are plausible and reiterate the importance of adequate hydration for an important aspect of cognitive function in older people.

The only suggestions I have is: i) Line 54: Sentence should read 'The majority of studies focusing on...'; and ii) that Table 2 be presented on the one page.

Changed.

Congratulations on your work and thank you for the opportunity to review such a well presented study report.

Thank you very much.